# Will Plant Genome Editing Play a Decisive Role in “Quantum-Leap” Improvements in Crop Yield to Feed an Increasing Global Human Population?

**DOI:** 10.3390/plants10081667

**Published:** 2021-08-13

**Authors:** Anton V. Buzdin, Maxim V. Patrushev, Eugene D. Sverdlov

**Affiliations:** 1The Laboratory of Clinical and Genomic Bioinformatics, I.M. Sechenov First Moscow State Medical University, 119991 Moscow, Russia; buzdin@oncobox.com; 2Moscow Institute of Physics and Technology, Dolgoprudny, Moscow Region, 141701 Moscow, Russia; 3Shemyakin-Ovchinnikov Institute of Bioorganic Chemistry, Russian Academy of Sciences, 117997 Moscow, Russia; 4Kurchatov Center for Genome Research, National Research Center Kurchatov Institute, 123182 Moscow, Russia; maxpatrushev@yandex.ru; 5Institute of Molecular Genetics, National Research Center Kurchatov Institute, 123182 Moscow, Russia

**Keywords:** CRISPR/Cas9, crop yield potential (Yp) limits, quantum leap, synthetic biology

## Abstract

Growing scientific evidence demonstrates unprecedented planetary-scale human impacts on the Earth’s system with a predicted threat to the existence of the terrestrial biosphere due to population increase, resource depletion, and pollution. Food systems account for 21–34% of global carbon dioxide (CO_2_) emissions. Over the past half-century, water and land-use changes have significantly impacted ecosystems, biogeochemical cycles, biodiversity, and climate. At the same time, food production is falling behind consumption, and global grain reserves are shrinking. Some predictions suggest that crop yields must approximately double by 2050 to adequately feed an increasing global population without a large expansion of crop area. To achieve this, “quantum-leap” improvements in crop cultivar productivity are needed within very narrow planetary boundaries of permissible environmental perturbations. Strategies for such a “quantum-leap” include mutation breeding and genetic engineering of known crop genome sequences. Synthetic biology makes it possible to synthesize DNA fragments of any desired sequence, and modern bioinformatics tools may hopefully provide an efficient way to identify targets for directed modification of selected genes responsible for known important agronomic traits. CRISPR/Cas9 is a new technology for incorporating seamless directed modifications into genomes; it is being widely investigated for its potential to enhance the efficiency of crop production. We consider the optimism associated with the new genetic technologies in terms of the complexity of most agronomic traits, especially crop yield potential (Yp) limits. We also discuss the possible directions of overcoming these limits and alternative ways of providing humanity with food without transgressing planetary boundaries. In conclusion, we support the long-debated idea that new technologies are unlikely to provide a rapidly growing population with significantly increased crop yield. Instead, we suggest that delicately balanced humane measures to limit its growth and the amount of food consumed per capita are highly desirable for the foreseeable future.

## 1. Introduction

Growing scientific evidence demonstrates unprecedented planetary-scale human impacts on the Earth’s system [1]. In 1971, John Harte published “*Patient Earth*” [2], which discusses rising problems in the nascent field of environmental science, including human population growth, resource scarcity, and nuclear contamination. In 1972, Donella Meadows and her colleagues published a landmark book entitled “*The Limits to Growth*” [3], the message of which is that the resources of the Earth probably cannot maintain the current rate of economic and population growth well after 2100 even with the use of advanced technologies, which is still in dispute today. The major challenges facing civilization have become undoubtedly evident [4]. Rising atmospheric CO_2_ levels were predicted by Svante Arrhenius in 1896 [1], and a strong correspondence between the temperature and the concentration of carbon dioxide in the atmosphere observed during the glacial cycles of the past several hundred thousand years can be seen at the site of NOAA’s National Centers for Environmental Information [5].

The warming atmosphere has influenced global wind and precipitation patterns and increased the intensity of extreme weather [6]. Catastrophic fires observed in different areas of the planet are evident expressions of this change. Atmospheric chemistry is also affected by human activities [7]. Food systems account for 21–34% of global emissions. Over the past half-century, land-use changes have significantly impacted ecosystems, biogeochemical cycles, biodiversity, and climate. The global area equipped for irrigation has doubled since the 1960s; agriculture now represents 70% of freshwater withdrawals around the world and global fertilizer use has quadrupled, leading to increased nutrient runoff into inland waters and coastal seas [8]. Expanding agricultural land use is a significant contributor to rising atmospheric CO_2_ levels and biodiversity loss due to deforestation and the draining of wetlands [9]. The list of influences could be continued. The causes of slowing global grain production and shrinking reserves [10] are yet to be answered [11]. Some predictions suggest that crop yields must double by 2050 to adequately feed an increasing global population without a large expansion of crop area, although this is a hotly debated issue [1,9,11,12,13,14,15,16,17]. Doubling agricultural yield within the next 30 years requires an annual increase of ~2.2%, which exceeds the average annual increase witnessed over the past 50 years [18]. We do not discuss the highly important problems of poor food distribution and wastage of one-third of the world’s food talked over in this review [19].

“Quantum-leap” improvements in crop cultivar productivity are needed to achieve sufficient annual yields within a narrow window of permissible environmental perturbations. In 2009, “the planetary boundaries framework” was put forward defining a “safe operating space for humanity” [20] (see also [21]). It was argued that a set of nine “planetary boundaries” must not be crossed by humanity at the cost of its own peril through scientifically defined targets of the maximum allowed human interference with processes that regulate the state of the planet. The nine processes are climate change; biogeochemical (nitrogen and phosphorus) flows; land-system change; freshwater use; aerosol loading; ozone depletion; ocean acidification; the loss of biosphere integrity, including functional and genetic biodiversity; and the introduction of novel entities, such as toxic chemicals and plastics. The concept of planetary boundaries has generated significant academic debate and policy recommendations worldwide [21,22,23]. However, it is a very useful reminder for people to be extremely cautious in transforming nature.

“Green Revolution” technologies of the post-war years have resulted in cereal production increases of 30% per capita over the last 50 years [8]. However, the revolution had great unintended consequences [24]. Greater than 90% losses in crop genetic diversity have occurred over the 20th century due to agricultural practices relying upon only three plant species: rice (*Oryza sativa*), maize (*Zea mays*), and wheat (*Triticum aestivum*) towards supplying nearly 60% of the world’s plant-based food [8]. Improvement in crop diversity is necessary to achieve resilience to abiotic and biotic stress [25]. Enhancement in crop yield and stability requires a systems approach by combining agronomic and technological management with the implementation of new crop cultivars [11,12,25,26]. Ideally, new crop varieties should have genetic combinations that alleviate losses from the multiple environmental and pest constraints encountered during the crop lifecycle in a farmer’s field. The integration of mechanistic understanding, genetic variation, and genome-scale breeding will be essential for technological solutions to manage shortfalls in agriculture yield and stability in a growing worldwide population [11,26,27]. This is also true for forage legumes [28], which play a crucial role as feedstock in the global production of meat [29] and occupy comparable planting space as food crops [30].

There are three strategies for significantly improving crop yield and stability: mutation breeding, genetic engineering, and directed modification. Mutation breeding involves exposing seeds to chemicals or radiation to generate mutants with desirable traits to be bred with other cultivars. Genetic engineering of crops with known genome sequences allows the creation of agronomically relevant variations [31], and synthetic biology techniques allow the synthesis of extended DNA fragments with desired sequences [32,33]. Currently, the most widely adopted genetically modified trait in crops is resistance to herbicides and insects, which has been incorporated into *Zea mays*, *Glycine max* (soybean), *Gossypium hirsutum* (cotton), and *Brassica napus* (canola). They are usually monogenic traits. Directed modification involves targeting specific DNA regions or selected genes that are known as important agronomic traits for DNA editing. New technologies such as “clustered regularly interspaced short palindromic repeats/CRISPR-associated protein 9” (CRISPR/Cas9) are being widely investigated for their potential to enhance crop production efficiency. These expectations are very high [15,16]. Bioinformatics plays an important role in the selection of targets for directed modification by relying on existing information about DNA, RNA, and protein sequences contained in databases such as GenBank, Ensembl, or UniProt as well as their functional activities contained in databases such as gene ontology (GO). A database Gramene for plants is being actively developed (see below, Section 12, page 15). However, most GO annotations are incomplete and imperfect [34,35]. Therefore, predicting the associations between genes and phenotypes is rather problematic, as is the identification of adequate targets for modification. It is suggested that a better understanding of the mechanisms controlling yields in variable environments is required for necessary crop improvement.

Here, we assess the validity of the optimism associated with new genetic technologies in terms of the complexity of most agronomic traits, especially crop yield potential (Yp) and its theoretical limits. We discuss possible directions for overcoming these limits and suggest alternative ways of providing humanity with sufficient food without transgressing planetary security boundaries. The suggestions vary from innovative urban agriculture development [36,37,38] to the development of crops tolerant to poor soil [39]. Due to the limited space, we do not consider the problems associated with the second and third green revolutions since these terms are heterogeneously interpreted by different authors.

## 2. The Green Revolution and Its Genes

Modern agriculture has its roots in the green revolution that began with the introduction of high-yielding wheat and rice cultivars in the 1960s [24,38,40,41,42]. Norman Borlaug, the father of the green revolution, bred wheat to favour shorter, stronger stalks that better support larger seed heads. In 1953, he crossed a Japanese dwarf variety of wheat called Norin 10 with a high-yielding American cultivar called Brevor 14. Norin 10/Brevor 14 is a semi-dwarf cultivar that is one-half to two-thirds the height of standard varieties and produces more stalks and, thus, more heads of grain per plant. Borlaug also crossbred the semi-dwarf Norin 10/Brevor 14 cultivar with disease-resistant cultivars to produce wheat varieties that are adapted to tropical and sub-tropical climates. Borlaug’s new semi-dwarf, disease-resistant varieties dramatically changed the potential yield of spring wheat. By 1963, 95% of Mexico’s wheat crops used the semi-dwarf varieties developed by Borlaug and the harvest was six times than that in 1944, the year Borlaug arrived in Mexico. Mexico has become fully self-sufficient in wheat production and a net exporter of wheat. 

Plant height is a major agronomic trait closely correlated with crop yield, which is controlled by multiple genes that may be optimized through breeding strategies (see “Complexity of agronomic traits” for further discussion). The genes responsible for dwarfing traits interfere with the action or production of gibberellic acid (GA) plant hormones. Two main “green revolution genes”, namely *Rht (**reduced height*), which encodes a growth repressor DELLA protein of GA signaling, and *sd1* (*semi-dwarf1*), which encodes (GA) 20-oxidase [41,43,44,45], were cloned from wheat and rice, respectively, and are now widespread through international breeding programs. Only 3 out of the more than 21 reduced-height (*Rht*) genes reported in wheat have been used extensively in wheat breeding programs (*Rht-1* homoeoalleles *Rht-B1b* and *Rht-D1b*, *Rht8*, and *Rht12*) [42,46,47,48,49]. Remarkably, both *Rht-1* homoeoalleles originate from the same Japanese variety, Norin 10. In addition to the widely used GA-insensitive dwarfing genes *Rht-B1* and *Rht-D1*, there is a wide spectrum of loci that can be used to modulate plant height [50]. Rice *sd1* was initially defined as a semi-dwarfism gene encoding a defective enzyme in the GA biosynthetic pathway [41,51,52,53,54], and pleiotropic changes were revealed by the recovery of the wild-type cultivar following restoration of the *sd1* mutant protein to wild-type in rice [54]. 

Since interference with the plant’s response to GA triggers adverse effects for a range of important agronomic traits, attempts have been made to identify mutants without these shortages. Recently, a major *Rht* locus on wheat chromosome 6A, *Rht24*, substantially reduced plant height alone as well as in combination with *Rht-1b* alleles. Unlike the two *Rht-1b* alleles, plants carrying *Rht24* remain sensitive to GA treatment [40]. Nowadays, as many as 61 genes (*d1* to *d61*) are known to cause dwarfism in rice [55].

In addition to GAs, brassinosteroids and strigolactones are also involved in controlling plant height. The genes involved in changing the levels of these hormones offer additional opportunities for expanding the genetic basis of semi-dwarf rice breeding [52,56]. The above-mentioned genes controlling plant height lie in a complex regulatory network, and additional dwarfing genes are involved in other pathways [40,56]. Phytohormones regulate many aspects of plant life by activating transcription factors that bind sequence-specific response elements (REs) in regulatory regions of target genes. Specific RE variants are highly conserved in core hormone response genes and regulate the magnitude and spatial profile of hormonal responses suggesting that hormone-regulated transcription factors bind a spectrum of REs, each coding for a distinct transcriptional response profile [57]. Such intricate regulation adds an additional level of trait complexity, see [58] for a detailed review.

In his 1970 Nobel lecture, Borlaug summarized the qualities of wheat that he had bred: “It is the unusual breadth of adaption combined with high genetic Yp, short straw, a strong responsiveness and high efficiency in the use of heavy doses of fertilizers, and *a broad spectrum of disease resistance* that has made the Mexican dwarf varieties a powerful catalyst that they have become in launching the green revolution” [59].

Plant diseases are responsible for substantial crop losses each year and pose a threat to global food security and agricultural sustainability. However, it is challenging to breed varieties with resistance that is effective, stable, and broad spectrum [60,61,62,63]. Plant growth and disease resistance are tightly regulated, and many negative correlations between growth and defence are the result of regulatory crosstalk [60]. There is increasing evidence that resistance to one disease involves trade-offs with responses to other bio-antagonists. For example, numerous pleiotropic effects of mildew resistance locus O (*MLO*) were recently reviewed [64]. There may occur substantial genotype-by-environment interactions in fitness costs, which makes experiments studying disease resistance in plants especially challenging [60,62,63,64,65,66,67,68].

Insecticide resistance mutations are widely assumed to carry fitness costs [66]. For example, an intercellular sucrose transporter was recently identified as a major susceptibility locus in blight-resistant rice *O. sativa* [69]. This transporter moves sugars from the photosynthetic tissues into the phloem for transport to tissues requiring externally supplied sugar for growth and development. A disadvantage of transporting sugars outside of the plant cell is the supply of a carbon source for endophytes [69]. Coordination between growth and disease resistance demonstrates the activity of GAs in the presence and absence of microbes [70,71]. When a microbe is detected, an immune cascade overrides the destabilising activity of GA on DELLA proteins and re-establishes DELLA-mediated suppression of growth [72].

It is suggested that genes with large effects on defence against a bio-antagonist also have large pleiotropic effects on survival, growth, reproduction, and responses to other bio-antagonists. The exceptions to this rule include many resistance genes but in many cases have little or no detectable fitness cost. Therefore, the alleles of resistance genes selected by plant breeders have extensive benefits in providing strong disease-resistant crops with yields comparable to wild-type [64].

Plants adopt various strategies to reduce the cost of mounting resistance. This largely involves the restriction of defences to particular tissues, developmental stages, and/or windows of time. Cost amelioration may involve tritrophic interactions [73,74], where the activity of another species (such as within a protective microbiome) prevents the invasion of pathogens [75].

The commonly accepted notion is that most spontaneous mutations are deleterious with negative fitness effects on the survival of the individuals who carry them, and only a small fraction is beneficial. It is also suggested that the majority of deleterious mutations have small fitness effects (1% or less on average) [76]. Meanwhile, compensatory mutations may counteract the negative effects of other deleterious mutations, although alone, they are also deleterious. A large variation in the fitness effects of deleterious mutations may be an important factor in the survival and growth of some small natural populations. *In silico* modelling of gene regulatory networks [77,78] implied that compensatory mutations are surprisingly frequent and can drive gene regulatory network evolution. Furthermore, predictions indicate that the smaller the population, the larger the effect of compensatory mutations on fitness recovery, with the compensatory effect increasing sharply with a decreasing population [76]. Future empirical studies should test this prediction, and, in any case, it should serve as a warning to researchers selecting favourable traits in experimental fields.

## 3. Theoretical Limits on Crop Productivity. A Complex System Cannot Be Predictably Modified but Can Be Replaced by a Functionally Similar Complex System

The theoretical limits of crop productivity were initially examined in 1992 by [79]. They stated that environmental limitations render crops unable to achieve their genetic yield potential even in the best field conditions. Crop growth in controlled hydroponic conditions with high CO_2_ levels is limited by photosynthetic photon flux even at daily levels that are three times higher than maximal summer sunlight. Therefore, biomass productivity and edible yield are still well below the predicted maximal output. Photosynthesis emerges as the key remaining route to increasing the genetic Yp of the major crops, yet it has improved little and falls far short of its biological limit. A better understanding of the mechanisms of photosynthetic processes should lead to the development of strategies for Yp improvement [80].

The Yp of crop can be described by the following equation [80,81,82]:Yp = Q⋅ε_i_⋅ε_c_⋅ε_p_

Q is the total solar radiation; ε_i_ is the efficiency of light capture; ε_c_ is the efficiency of conversion of the intercepted light into biomass; and ε_p_ is the harvest index, the proportion of biomass partitioned into grain. In the absence of environmental stress, parameters, such as harvest index, are close to the theoretical maximum. Plant breeding brings ε_p_ and ε_i_ close to their theoretical maxima, leaving ε_c_, primarily determined by photosynthesis, as the only remaining major prospect for improving Yp [82]. This is an excellent idea, and it has many active apologists.

Photosynthesis is dependent upon the interactions between chloroplast organelles and the nucleus [32,83] since chloroplast fitness relies on nuclear-encoded genes. In angiosperms, the chloroplast genome (plastome) expresses only approximately 50 protein-coding genes involved in tRNA and rRNA genes, chlorophyll synthesis, photosynthesis, and metabolic processes [84]. Meanwhile, up to 2500–3500 nuclear-encoded proteins are predicted to be chloroplast localised in *Arabidopsis thaliana* [84]. However, organelle-to-nucleus signaling coordinates the expression of nuclear genes encoding chloroplast proteins with the metabolic and developmental state of the organelle [85].

Improvement of the currently low conversion efficiency of light to biomass (~2%) has received considerable attention [86]. The dramatic environmental changes that land plants have repeatedly experienced in the course of their evolution probably resulted in the formation of a rather robust photosynthetic system, with unoptimised efficiency for particular conditions. This means that modern photosynthetic productivity is maintained at optimal rates under many adverse conditions rather than tuning efficiency to conditions used in modern farming [87]. It is predicted that improving photosynthetic efficiency will not be easy. The successful modification of photosynthesis to enhance plant growth and yield has been limited to a few cases [88,89]. This provides some support that the genetic engineering approach is an avenue worth pursuing for the improvement in Yp through the optimisation of photosynthetic processes [18,80,81,90,91].

Four major research areas for redesigning photosynthesis were suggested [32]: (i) studying natural variations in photosynthesis, (ii) coordinating photosynthesis with pathways using photosynthesis, (iii) transfer of highly efficient photosynthetic systems existing in non-host species, and (iv) engineering photosynthetic systems not existing in nature.

Overexpressing *A. thaliana* SBPase in tobacco (*Nicotiana tabacum*) was the first successful photosynthetic carbon metabolism engineering. The transgenic plant had higher SBPase activity, increased photosynthetic rate, greater accumulation of sucrose and starch, and a higher total biomass increment (for a brief recent review see [83]); we also mention a couple of other examples reviewed in [83] for clarity.

SBPase activity was increased in the transgenic rice cultivar (*Oryza sativa* L. ssp. *japonica*) by overexpressing OsSbp cDNA from the rice cultivar 9311 (*Oryza sativa* ssp. *indica*). The transgenic plants accumulated SBPase in chloroplasts and developed enhanced tolerance of transgenic rice plants to salt stress at the young seedling stage [92]. Chilling is a factor limiting growth and yield in tomato production. Genetically engineering tomato plants with an appropriate target gene could ameliorate the chilling injury. It was reported [93] that in transgenic tomato plants (*Solanum lycopersicum*) with increased SBPase activity, photosynthetic rates were increased as well as sucrose and starch accumulation. Tomato plants with increased SBPase activity were more chilling tolerant. Thus, the higher level of SBPase activity provides an advantage to photosynthesis, growth, and chilling tolerance in tomato plants. Another consequence of this work is that an individual enzyme in the Calvin cycle may be a useful target for genetic engineering to improve production and stress tolerance in crops.

The level of the SBPase in wheat has been increased through transformation and expression of a *Brachypodium distachyon* SBPase gene construct and showed enhanced leaf photosynthesis and increased total biomass and dry seed yield [94]. South et al. inserted a synthetic glycolic acid metabolic pathway in *N. tabacum* chloroplasts by expressing pumpkin (*Cucurbita maxima*) malate synthase and green alga (*Chlamydomonas reinhardtii*) glycolate dehydrogenase and demonstrated that engineering alternative glycolate metabolic pathways into crop chloroplasts while inhibiting glycolate export into the native pathway can drive increases in C3 crop yield under agricultural field conditions [95].

These examples show that challenges in Yp improvement can potentially be overcome using genetic engineering in conjunction with synthetic biology and computational modelling strategies [80]. Synthetic biology tools allow the redesign of entire processes using simpler existing systems, for example, by introducing algal/bacterial ‘inventions’, such as carboxysomes, into land plants [87]. Moreover, instead of changing single components, synthetic biology tools allow the engineering and redesigning of entire processes [18,83,89,90,91].

On the other hand, we can expect a certain success from the side of “blue” biotechnology, namely from unicellular algal biotechnology. Algae are important sources of nutrients for both humans and agricultural species [96,97]. Microalgae can produce biomass that is enriched in proteins [98,99,100], low saturated fatty acids [101], and antioxidants [102]. Since they have comparatively rapid growth rates with varying photosynthetic apparatus and mechanisms [103], they may be regarded as attractive targets for genetic modifications to improve their metabolic productivity. However, scepticism remains as to whether increased photosynthetic capacity may increase food crop yields [104] (see below).

## 4. Complexity of Agronomic Traits

The cause of slowing crop yield growth is yet to be determined [11]. The gap between the actual yield and yield potential may be accounted for by several variables, including genetics, environment, management practices, and socioeconomic factors. These results indicate that many agronomic traits in plants are complex systems. Complex genetic architectures include numerous interacting loci (or alleles) with small effects and interactions with the genetic background, environment, or age. Complex agronomic traits, such as plant height, harvest index, total biomass, number of productive tillers, grain number per spike, spike length (SL), number of kernels per spike, thousand seed weight, and grain weight per spike, and physiological traits, such as canopy temperature (CT), chlorophyll content, photosynthetic rate, and water-soluble carbohydrates (WSC), contribute to grain yield improvement in wheat [105]. The complexity of optimizing a particular complex trait may be visualised by a comparison with recent research demonstrating that human height is associated with at least 10,000 DNA markers [106] and missing heritability [107].

Yield depends on many intrinsic and extrinsic factors, such as plant height, biotic and abiotic stresses, the efficiency of light energy capture by photosynthetic light reactions, the efficiency of conversion of light energy into biomass, and harvest index [18]. Such complex traits are defined as quantitative traits (QT). These traits are collectively regulated by several loci (QTL) that may interact with each other and with the environment and affect the mode of gene action. Recent advances in next-generation sequencing make it possible to map QTL genomic regions and help map phenotypes for thousands of traits. This leads to the partial reconstruction of gene networks at the transcript level and explains the relationship among traits [55,108,109]. Many agronomically important traits are quantitatively inherited, especially yield and yield-contributing traits [49,55]. Crop yield is a QT [55] that is controlled by many plant genes. In wheat, for example, three main phenotypic yield components were identified: thousand kernel weight (TKW); kernel number per spike; and spike number per unit area (SN), which determines wheat yield. Correspondingly, many genetic loci related to wheat yield have been identified. Recently, 58 QTL-rich clusters were defined based on their distribution on chromosomes [110]; however, their complete genetic architecture and key genetic loci for selection remain largely unknown [110]. Current methodologies in quantitative genetics [111] can only detect the determinant genes with the strongest effects, which are unlikely to represent all of the components required to produce a phenotypic characteristic [112]. There is every reason to believe that there are constraints on the magnitude of allowable variations of regulatory genes that are typically dosage sensitive. This multifactoriality makes it difficult to identify the appropriate targets for gene editing.

Recently, the role of the environment in the variability of phenotypic traits in maize, including crop yield, was investigated [113]. The variation in observed phenotypes can be partitioned into three main factors: genotype, environment, and genotype × environment interaction (G × E), in addition to other minor factors and measurement errors. In plant breeding, G × E plays an important role as the relative performance of different genotypes in different environments influences plant breeders’ recommendations of best-performing varieties for specific regions. Typically, plant breeders minimize G × E by producing cultivars that are appropriate for regions that share common environmental characteristics. With an improved understanding of specific components of genotype, environment, and G × E, breeders may use data-based approaches to enhance their ability to position a larger number of genotypes in environments to maximize productivity. Grain yield is of primary importance in breeding *Zea mays* L. and is commonly considered alongside several traits that affect it indirectly or directly, such as flowering, height, and yield-component traits. Due to their differences in heritability and sensitivity to environmental factors, these different types of traits may show different levels of G × E [113]. G × E interactions showed between 9.0 and 20.4% of the phenotypic variance with greater effects in the yield-component traits [113].

However, the ability to fine-tune the expression of a QTL rather than only utilizing what is available in wild relatives is shown as a promising way to increase yield. For example, researchers edited genes that altered the promoters in three pathways contributing to productivity in tomato plants—plant architecture, fruit size, and inflorescence [11]. The resulting plants displayed a series of previously unobserved phenotypes, including several with increased yields.

## 5. Nitrogen Input

In the current bottleneck of crop production, we should leave habitual standards and search for new approaches to the problem of human survival. The considerations of experienced sceptics should be considered, for example, T. R Sinclair et al. [104]: “It seems crucial to further elucidate the role of resource inputs other than carbon in influencing crop yield... Given the conclusion that nitrogen input to crops has been and will continue to be critical in limiting crop grain yield, there are important questions for future research targeting nitrogen availability and use in crop plants…”.

In the above-cited Nobel lecture, Borlaug stated: “In my dream I see green, vigorous, high-yielding fields of wheat, rice, maize, sorghums, and millets, which are obtaining, free of expense, 100 kg of nitrogen per hectare from nodule-forming, nitrogen-fixing bacteria...”.

In the following sections, we discuss some of the strategies directed at the solution of this critical issue. 

## 6. Is It Possible to Transfer Nitrogen-Fixing Genes from Legumes to Non-Legumes?

Biological N_2_ fixation, catalysed by the prokaryotic enzyme nitrogenase, is an attractive alternative to the use of synthetic N fertilizers. Associations with nitrogen-fixing bacteria delivering the complete nitrogen needs of the host plants are limited to a select group of species. It is tempting to try to radically improve nitrogen availability for cereal crops by transferring the symbiosis trait encoding genes from legumes to non-legumes, especially to economically important crops, such as rice, maize, and wheat (reviewed in [25,114]).

Extensive genetic and biochemical studies have identified the common core set of genes/gene products required for functional nitrogen biosynthesis [25,114,115,116]. Molybdenum nitrogenase is an O_2_-labile metalloenzyme composed of NifDK and NifH proteins, which requires several *nif* gene products. The sensitivity of nitrogenase to O2 and the apparent complexity of nitrogenase biosynthesis are the main barriers identified to date. The expression of active NifH requires NifM and NifH and possibly NifU and NifS, whereas active NifDK requires NifH, NifD, NifK, NifB, NifE, and NifN and possibly NifU, NifS, NifQ, NifV, NifY, NifW, and NifZ. Plastids and mitochondria are potentially viable subcellular locations for nitrogenases since they provide the ATP and electrons required for nitrogenase activity. These organelles differ in their internal O_2_ levels and their ability to incorporate ammonium into amino acids. The direct transfer of *nif* genes into cereals to increase cereal crop productivity is challenging due to the sensitivity of nitrogenase to O_2_ and the apparent complexity of nitrogenase biosynthesis.

Associations with nitrogen-fixing bacteria that deliver the complete nitrogen needs of the host plants are limited to a select group of species. The ability to fix nitrogen has been found in a wide range of bacterial genera, many of which are known to be associative (residing on or near the root surface) and endophytic (residing within plant cells) rhizobacteria, including *Azospirillum*, *Azotobacter*, *Burkholderia*, *Gluconacetobacter*, *Herbaspirillum*, *Klebsiella*, *Paenibacillus*, and *Pseudomonas* [117]. Engineering biological nitrogen fixation in plants by the direct introduction of *nif* genes, as well as in the case of photosynthesis, requires elegant synthetic biology approaches to ensure highly active and stable nitrogenase activity through expression in the appropriate stoichiometry of the subunits. This, if possible, may be achieved by synthetic engineering of the nitrogenase system into mitochondria or chloroplasts since these organelles potentially provide the reducing power and ATP required for nitrogen fixation [118].

## 7. Taking Advantage of Plant–Microbe Interactions

Billions of microorganisms and macroorganisms (from viruses to nematodes) live on, inside, and near plants, both above and below ground [119]. The results of beneficial plant–microbe interactions include the direct stimulation of plant growth, the protection of plants from pathogens and insect pests through the direct production of toxins or through induced resistance in the plant, and improved resilience to environmental stress (e.g., drought, salinity). Beneficial interactions occur in the root zone (rhizosphere), leaf surfaces (phyllosphere), and internal tissues (endosphere) [11,120,121]. In leguminous plants, some *Rhizobium* species of bacteria induce the formation of root nodules in a symbiotic relationship that converts atmospheric and largely inert N_2_ into ammonia (NH_3_) and other molecular precursors that the plant uses in the biosynthesis of nucleotides, coenzymes, and amino acids. In many more species of plants, fungal symbionts (arbuscular mycorrhizal fungi) form hyphae that increase the ability of plant roots to access minerals (particularly phosphorus) and water [119]. The hundreds of land plant and algal genomes that are now available enable genome-wide comparisons of gene families associated with plant immunity and symbiosis. However, few plant–microbe interactions have been studied in depth, with only a few land plant lineages. Subsequent investigations may reveal new types of symbiotic or pathogenic interactions [117,122]. Synthetic biology tools may provide an opportunity to design plant–microbe associations that improve crop productivity. Such associations can be studied using gene editing in plants, microbes, or both [118]. For example, plant genes controlling nodule formation by nitrogen-fixing rhizobacteria may be expanded to non-legume crops to reduce the need for fertilizer application, and microbial consortia present in the root zone could be engineered to produce novel plant growth promoters or protectants.

However, the complexity of plant genetics, metabolism, and nitrogen fixation machinery makes this an extremely challenging task [117,122]. Early ancestral cereals were associated with nitrogen-fixing bacteria according to a recent study (reviewed in [117]). Efforts are being made by multiple research groups to enhance biological nitrogen fixation in cereal crops through (a) enabling symbiosis between plants and nitrogen-fixing bacteria using genetic engineering and (b) identifying and utilising nitrogen-fixing bacteria to supplement plant nitrogen requirements. Symbiosis of native cereal crops with root-associated nitrogen-fixing bacteria offers a potentially sustainable solution for nitrogen management on a shorter timescale [114,117,123].

Utilising the plant microbiome is a reliable approach for the next green revolution and to meet global food demand in sustainable and eco-friendly agriculture [124]. Technological developments such as next-generation sequencing, gene editing, and synthetic biology allow the manipulation of plant and microbe genotypes at an unprecedented scale. Combining the prospecting of plant and bacterial natural diversity with genetic engineering will hopefully provide a more sustainable global food source in the short and long term [123].

## 8. Plant-Mediated Strategies for Shaping the Rhizosphere Microbiome

Characteristics of interest are manipulated using two different approaches: plant breeding and genetic engineering (see [121] and references therein). Plant breeding techniques for selecting a specific microbial community aim to increase crop yield by providing plant resistance to a variety of stresses. Many studies have manipulated plants by modifying the production of key exudates, which directs the establishment of specific plant microbiomes. For example, Koyama et al. [125] reported that transgenic plants have a greater ability to secrete citrate from the roots and therefore grow better in phosphate-limited soil. Yang et al. [126] and Gevaudant et al. [127] manipulated the pH of the rhizosphere by using transgenic lines of *Arabidopsis thaliana* and *N. tabacum* plants overexpressing an H^+^-ATPase protein, which increased H^+^-efflux from the roots of the plant. This created a more acidic environment in the rhizosphere resulting in enhanced growth at a lower pH, increased resistance to drought stress through the expression of pyrophosphate-energised vacuolar membrane proton pump 1 (AVP1), and augmented tolerance to salinity stress in tobacco lines [128]. Similar examples can be continued. More recently, site-directed genetic engineering of DNA has used methods such as CRISPR/Cas9 and transcription activator-like effector nucleases (TALENs).

## 9. Genome Editing Technologies for “Quantum-Leap” Improvements in Yield-Limited Crops Are Ready, but Where Are the Targets?

There are two problems in targeted genome editing: (1) the identification of target(s), which should be modified to achieve desirable phenotypic change, and (2) precise targeted genome modification. The advent of precise genome-editing tools is expected to revolutionise the way we create new plant varieties. Three groups of tools currently available are classified according to their mechanism of action: programmable sequence-specific nucleases, base-editing enzymes, and oligonucleotide-directed mutagenesis [129,130]. The most commonly used today is CRISPR/Cas9, which has been implemented in more than 20 crop plants (reviewed [121,130,131]) for a variety of desired traits to improve crop yield and management of abiotic and biotic stress tolerance in plants. This genome-editing technology is adopted from the prokaryotic adaptive immunity system, which is found in several bacterial and archaeal genomes. It uses small stretches of RNA sequences coupled with nucleases Cas-, the enzymes that specifically cut the genome of invading viruses to suppress them. This system is used for the introduction of desired variations at the chosen location in the genome [132]. The technology can be applied to modify virtually any genomic sequence with the only restraint being the accessibility of the protospacer adjacent motif (PAM). PAM is a short, typically 2–6 bp sequence recognised by any compatible Cas-nuclease sequence near the target sites. Different Cas-nucleases distinguish various PAMs, and, frequently, there are even differences among orthologs. In such a way, site-directed nucleases (SDNs) are constructed and used for targeted genome editing to introduce double-stranded breaks (DSBs) at precise sites into plant genomes. This technology can be used to modify virtually any genomic sequence with the only restraint being the accessibility of the PAM near the target sites [133,134].

In the immense literature devoted to the CRISPR technology used for editing plant genomes, we found 39,100 articles with the words “CRISPR plant editing”, which contained contradictory data on the efficiency and specificity of this technology. As this technology is widely used, we do not describe it in detail here since it is best described in multiple reviews [15,16,132,135,136,137,138]. CRISPR/Cas9 is economical, easy to use, highly accurate, and effective even when performing multiplex genome editing (MGE) [15,139,140,141,142,143,144]. CRISPR/Cas9, CRISPR/Cas12a, and base editing are being improved continuously [15]. A simple scheme showing that this technology may be used to insert point mutations or extended DNA fragments into defined genomic sites is shown in Figure 1.

MGE involves the simultaneous targeting of multiple related or unrelated targets. The latter is the most straightforward using the CRISPR/Cas9 system because multiple guide RNAs (gRNAs) can be delivered either as independent expression cassettes with their own promoters or as polycistronic transcripts processed into mature gRNAs by endogenous or introduced nucleases. MGE in plants initially focused on input traits, such as herbicide resistance, but has recently expanded to include hormone biosynthesis and perception, metabolic engineering, plant development, and molecular farming, with numerous simultaneous targeting events reported. Knockout mutations in all three homologs of TaMLO (*T. aestivum* mildew resistance locus o) provided resistance against powdery mildew in wheat [141]. A non-complete list of edited genomes can be found [145] (see above, “Complexity of agronomic traits”).

Direct modification typically involves targeting protein-coding regions; however, recent examples include promoter modifications to generate mutants with varying gene expression levels [11,142,146]. CRISPR/Cas9 editing of promoters generates diverse cis-regulatory alleles that provide beneficial quantitative variation for breeding. A genetic scheme was devised to rapidly evaluate the phenotypic impact of numerous promoter variants on genes regulating three major productivity traits in tomato: fruit size, inflorescence branching, and plant architecture. This procedure allows for the immediate selection and fixation of novel alleles. It also provides an approach for testing complex relationships between gene regulatory changes and the control of QTs [147].

Researchers have paid great attention to the frequency of nonspecific mutations in CRISPR/Cas9. The data are highly contradictory, with one report stating an off-target specificity of 9.8–97.3% in *Arabidopsis* [148] to no evidence of off-target cleavage activity when specific gRNAs predicted by bioinformatics were chosen [149]. DNA breaks introduced by single-gRNA/Cas9 frequently resolved into deletions extending over many kilobases [150]. Furthermore, lesions distal to the cut site and crossover events have been identified [150]. Specificity may depend on the delivery method [151], and the problem of off-targeting may be tackled by the use of recently discovered CRISPR/Cpf12a (Cpf1), which creates a staggered double-strand break at the target site [136,152]. However, recently, Murugan et al. [153] revealed that Cas12a has multiple nicking activities against dsDNA substrates. SDN-mediated off-target changes can contribute to a small number of additional genetic variants compared to those that occur naturally in breeding populations or are introduced by induced-mutagenesis methods [154].

The second modified CRISPR “base editor” system can generate precise single-base mutations in the targeted DNA. It does not rely on DSB formation to induce targeted changes but instead uses a partly disabled nuclease with an additional protein domain. The targeting components of the nucleases are still intact, thereby allowing site-directed nucleotide changes and additional protein units to target specific genomic locations. Applications include targeted base editing with deaminase domains, transcriptional knockdown using repressors, targeted DNA methylation, and numerous other applications. In addition, the technology does not need the introduction of DSB to modify a base pair, and, consequently, the likelihood of major perturbations in the genome, such as deletion or chromosomal translocations, is considerably reduced. For example, cytosine and adenine base editors converting C to T and A to G, respectively, fuse a nickase-type Cas9 with a deaminase domain and, thus, do not induce DSBs (Figure 2). The two single-base editors (cytidine deaminase and adenosine deaminase) were fused to produce simultaneous double-base conversions (C → T and G → A) [15,154,155,156,157,158,159].

Cas9 can be easily adapted to facilitate genome-scale perturbations. For example, Cas9 nuclease can be converted into an RNA-guided DNA-binding protein (dCas9) and then fused to transcription activation or repressor domains. These dCas9-activator fusions target the promoter/enhancer regions of endogenous genes to modulate gene expression [160,161].

The significance of nonspecific CRISPR-caused mutations is unclear given the natural background of mutations that constantly appear stochastically in the genome. The estimated haploid spontaneous single nucleotide mutation rate for *A. thaliana* is about 7 × 10^−9^ per site per generation [162]. Approximately the same rate was reported in rice lines with 3.4-fold higher mutation rates in heterozygotes (1.1 × 10^−8^) than homozygotes [163]. The spontaneous mutation rate in *Zea mays* is 2.2 to 3.9 × 10^−8^ per site per generation [164] (reviewed in [154]). A detailed analysis of various off-target effects is presented in an excellent review by Graham [154]. Other stochastic de novo mutations occur during in vitro culture referred to as somaclonal variation. The estimated mutation rate in *Arabidopsis* root explants (living cells transferred to culture medium) is between 4 × 10^−7^ and 2.4 × 10^−6^ mutations per nucleotide, while a mutation rate of 1.0 × 10^−7^ occurs in rice plants regenerated through tissue culture [154]. During in vitro culture, many regenerated plants develop differences in appearance relative to the parental genotype, and these induced changes may include heritable genetic and epigenetic alterations [154].

When discussing the significance of off-target mutations, it should be noted that plants differ from animals in substantive ways that alter the impact of induced changes. First, unlike many animals, genetic changes in juvenile plants can be transmitted to reproductive tissues [154]. In addition, plants frequently develop multiple independent reproductive structures, with only a fraction affected by new mutations. Breeding to develop new lines for commercial release involves an intensive process of selection of individual plants with useful phenotypes while eliminating individuals with undesirable mutations or phenotypes (commonly known as “off-types”) [154]. For these reasons, off-target edits in crops present fewer safety concerns than those that could arise with the therapeutic applications of genome editing.

In conclusion, it should be mentioned that to generate a wide variety of new traits in plants using CRISP/Cas genome editing [165], Agrobacterium-, biolistic-, and also virus-mediated methods were used to deliver CRISPR/Cas into plant cells [166,167]. A high-throughput gene-editing assay Automated Protoplast Transformation System was recently developed [168].

## 10. Success Stories of CRISPR/Cas9

Since the introduction of CRISPR/Cas9 for editing mammalian genomes, it has been applied to modify the genomes of several model and crop plants, including tobacco, tomato, barley, *Arabidopsis*, wheat, rice, and maize. The genes involved in the regulation of fruit size and signalling pathways have been suggested as promising targets for genome editing-based crop improvement [169]. Following the first reports of CRISPR/Cas9-based genome editing in wheat protoplasts, gene-edited plants were generated using transformation with CRISPR/Cas9 in different forms, including plasmids, linear DNA fragments, linear RNA, and ribonucleoprotein complexes (reviewed in [139,140,141]).

Considerable efforts have been undertaken to identify QTLs controlling yield in various crop plants [15,165]. This was achieved in a rice cultivar by individually knocking out four negative regulators of yield (*Gn1a*, *DEP1*, *GS3*, and *IPA1*) using CRISPR/Cas9. Three of the resulting knockout mutations (*Gn1a*, *DEP1*, and *GS3*) showed enhanced yield in the T2 generation with increased grain size and number and denser erect panicles with a 30–68% increase in yield per panicle (reviewed in [15,165]). The editing of two yield-regulating genes, *Gn1a* and *DEP1*, developed superior alleles in rice with even greater yields than those of the natural high-yield alleles [170]. Similarly, the simultaneous knockout of three major rice negative regulators of grain weight (GW2, GW5, and TGW6) using a CRISPR/Cas9-mediated MGE system resulted in a significant increase in the thousand-grain weight. This approach can be used for the rapid breeding of QTLs in crop varieties [144].

Several phytohormones, such as abscisic acid (ABA), control plant growth and stress responses that affect crop yields. CRISPR/Cas9 generated mutations in genes encoding the ABA receptors pyrabactin resistance 1-like 1 (PYL1), PYL4, and PYL6 and created a rice line that produced up to 31% more grains than the original variety in field tests. This work highlights the potential of modifying hormones to control growth and improve yields in rice [171] (reviewed in [15]).

The successful modification of a gene encoding a maize negative regulator of ethylene responses, *ARGOS8*, using CRISPR/Cas9 was reported (reviewed in [165]). The homology-directed repair pathway was used to insert the maize native *GOS2* promoter into the untranslated region of *ARGOS8*, resulting in drought-tolerant maize with improved yield in limited water supply. Other studies reviewed in [165] confirmed that CRISPR/Cas9 can be used to manipulate abiotic stress genes, indicating its potential for future crop improvement. However, several essential traits, such as crop yield and abiotic stress resistance, are controlled by multiple genes, and the same QTL can have highly varied and opposing effects in different backgrounds.

These positive examples are certainly impressive; however, one should expect that they are the exception rather than the rule when a complex trait is the target for modification. For complex traits, such as crop yield, the result is expected to be unpredictable. Although CRISPR/Cas9 mutagenesis has improved our options for addressing gene function, recent results suggest that compensatory mechanisms in CRISPR mutants may hide gene functions [77]. Possibly, there may be no revolutionary breakthrough in plant selection by using directed mutations or changes in the genes involved in the formation of a complex trait, such as crop yield [105]. On the other hand, CRISPR/Cas9 mutagenesis has the unique capacity of seamless mutation by directly changing certain nucleotide positions without affecting the backbone sequence. This offers new possibilities and challenges to plant biotechnology companies and society and, theoretically, can result in an accelerated evolution of genetically modified organism (GMO) crop species that cannot be identified using traditional algorithms. In turn, surpassing the regulation of “hidden” GMO consumption may result in dramatic economical, ethical, and biotechnological implications.

## 11. Reference Genomes and Assessment of Genomic Variation

A prerequisite for target identification is the availability of a comprehensive and reliable sequence of the genome and its functional map. Although complete sequences of the genomes of higher plants are not well studied in comparison with the genomes of humans and animals, there has been rapid progress in this field from a highly fragmented genome assembly with incomplete gene models to a full “pseudomolecule” reference sequence along with detailed gene model annotation. Reference sequence allows the physical anchoring of genes in complete chromosomal order and provides improved gene models that facilitate the design of transgenic constructs and primers [172,173].

Many of the crop and vegetable species that constitute a major part of the global diet now have high-quality reference genome sequences (reviewed in [154,172]). However, reference genomes have several limitations, the most apparent being that no genes or gene variants are present in any single accession. The steps required to assemble pan- and super-pangenomes were reviewed in [174,175,176,177]. It is important to note that most whole-genome sequencing studies to date have used short-read sequencing technologies. As a result, the diversity in breeding populations due to structural variations, such as differing transposable element location and abundance, presence–absence variation (PAV), and gene copy number variants (CNVs), have been difficult to measure. PAVs and CNVs typically refer to changes that include genes. Although structural variants are less common than single nucleotide polymorphisms (SNPs), they are an important source of variation. There are many examples of gene PAVs or CNVs impacting agronomic traits (reviewed in [154]). Due to the density of naturally occurring variation, intra- and interspecific crosses of plants that occur during plant breeding of millions of SNPs and thousands of PAV or CNV sequence variations appeared. More long-read sequencing technologies will allow more accurate measurements of polymorphisms in breeding populations [154].

Plant genomes are notoriously repetitive and difficult to assemble [178], though long-read sequencing technologies have been quickly adopted [178,179] allowing high-quality de novo assembly. The rapidly increasing number of long-read, whole-genome sequencing (WGS) produces an increasing number of high-quality plant genome assemblies.

## 12. The Research Bottleneck in Plant Sciences Is Shifting from Genotyping to Phenotyping

Despite the tremendous progress made with continually expanding genomic technologies to unravel and understand crop genomes, the impact of genomics data on crop improvement is still far from satisfactory. This is largely due to a lack of effective phenotypic data and problems with genome functional mapping. Our ability to collect high-quality phenotypic data lags behind the current capacity to generate high-throughput genomics data. Thus, the research bottleneck in plant sciences is shifting from genotyping to phenotyping for unlocking information coded in plant genomes. The phenomics data generated have been used to identify genes/QTLs through QTL mapping, association mapping, and genome-wide association studies (GWAS, see below) for genomic-assisted breeding for crop improvement [47,180,181,182].

Bioinformatics, which is known to play an important role in the selection of targets for targeted modification, relies on existing information about the DNA, RNA, and protein sequences contained in databases such as GO. In 1998, the GO consortium released the first common vocabulary describing gene function across species, thus enabling a genome-wide and comparative approach to functional genomics [183]. The current release (10 August 2020) has 44,262 GO terms, 8,047,076 annotations, 1,556,208 gene products, and 4643 species [184].

A knowledgebase Gramene [185] based on the comparative functional analyses of genomic and pathway data for model plants and major crops contains, in the current release, 93 reference genomes over 3.9 million genes in 122,947 families with orthologous and paralogous classifications. Gramene integrates ontology-based protein structure–function annotation and information on genetic, epigenetic, expression, and phenotypic diversity; gene functional annotations extracted from this latest achievement will undoubtedly play an important role in the functional mapping of plant genomes.

Genomic databases have been powerful in integrating data from multiple studies, and international efforts are now bringing together phenotypic data alongside genotypic data (e.g., [186,187,188,189]. However, most GO annotations are incomplete and imperfect [34,35,190,191]. This is also true for the Gene Ontology Meta Annotator for Plants (GOMAP), which is an optimised, high-throughput, and reproducible pipeline for genome-scale GO annotation for plant genomes [189]. Therefore, predicting the associations between genes and phenotypes is rather problematic, as is the identification of adequate targets for modification. Several other challenges remain, with the most common for crop species being polyploidy, which is particularly evident in wheat. Due to functional redundancy, it will be necessary to knock out all homoeologs of a gene to determine its phenotypic impact [172,173,192]. In wheat, over 98% of the genome is non-coding; therefore, it is necessary to identify open chromatin regions to define non-coding but functionally important regions. Finally, it is essential to compare multiple wheat varieties to observe the effects of the same editing in different genetic backgrounds.

GWAS is an approach used in genetic research to associate specific genetic variations with phenotypic traits [34,35,190,191]. The method involves scanning the genomes of many different individuals and searching for genetic markers that can be used to predict the presence of a trait. The genetic markers can be used to understand how genes contribute to the trait and to uncover causal genetic polymorphisms in plants. This will aid breeders in developing improved plant varieties to meet the food needs of an ever-increasing world population.

Genome-wide screenings were first applied to humans a few years before the methods were leveraged for use in plants. In 2004, a publication first appeared that applied GWAS-like methods to barley (reviewed in [193]). GWAS is a good first step towards the discovery and deployment of key genes, further research is necessary to evaluate the reproducibility and transferability of GWAS results across environments and genetic backgrounds. The development of optimal experimental settings for GWAS analysis will require an interdisciplinary approach. The identification of key traits involves GWAS, proper analytical methods, using appropriate genetic resources for mapping, and choosing an adequate genotyping platform. With the arrival of rapid genotyping and next-generation sequencing technologies, GWAS has become a routine strategy for decoding genotype–phenotype associations in many species. Over the last decade, more than 1000 studies have revealed substantial genotype–phenotype associations in crops and provided opportunities to probe functional genomics [193,194,195]. A successful GWAS should to incorporate elements of candidate gene discovery and QTL deployment. Therefore, the following criteria for a successful GWAS were proposed: the study needs to identify true marker-trait associations with meaningful effect sizes; proximity to underlying genes for the traits of interest; and transferability within a similar population and across a reasonably broad set of environments [193].

## 13. The Use of Locally Available Resources to Adapt to Climate Variability and Change

A quite reasonable strategy for improving crop yield is the use of locally available resources to adapt to climate variability and change [196]. There are more than 50,000 edible plants, but only 15 crops are used, resulting in 90% of the world’s demand, and three of them (rice, maize, and wheat) provide two-thirds of human caloric intake [197]. Ironically, more than 70% of wild relatives of domesticated crops are threatened by extinction and in urgent need of conservation due to the expansion of agriculture into natural ecosystems [198]. Globally, gene banks and botanical gardens hold more than 7.4 million seeds or plant tissues from thousands of species [197]. These collections must be maintained, curated, and explored [11]. Among the most promising candidates are orphan crops that have either originated in a geographic location or those that have become ‘indigenised’ over many years of cultivation; they may offer ‘new’ opportunities as they are uniquely suited to harsh local environments, provide nutritional diversity, and enhance agrobiodiversity within farmer fields [199]. There is quite a lot of space to improve orphan crops, and genome editing accelerates modifications that would be problematic in a traditional breeding program. [200,201].

## 14. De Novo Domestication of Wild Plants

An attractive alternative route for future agriculture is the de novo domestication of wild plants [11,200,202,203]. This involves a multidisciplinary approach, including research from botany, archaeology, genetics, biogeography, and other disciplines [204]. The transformation of wild plants into domesticated crops usually involves modification of a common set of characteristics across different species, referred to as ‘domestication syndrome’ traits and previously defined as “the characteristic collection of phenotypic traits associated with the genetic change to a domesticated form of an organism from a wild progenitor form” [205,206]. Examples of these traits include loss of pod shattering/seed dehiscence, loss of seed dormancy, reduced anti-nutritional compounds, changes in growth habit, phenology, flower colour, and seed colour. Understanding the genetic control of domestication syndrome traits facilitates the efficient transfer of useful traits from wild progenitors into crops through crossing and selection [205,207]. It has become apparent in recent years that understanding the nature of the plurality of processes underlying domestication syndrome is the key to understanding the origins of domestication.

## 15. Reference, Pan-, and Super-Pan Genome Sequences Provide a Strong Basis for the Location of Domestication Syndrome Genes

In recent years, there has been a large influx of plant genome sequencing projects. The high level of genomic variation led to the realisation that single reference genomes do not represent the diversity within a species and led to the expansion of the pangenome concept. This suggests that the genomes of individuals within a population or species share a core set of genes that unifies them (the core genome) but also contains a fraction of genes that are absent from one or more individuals (the accessory or dispensable genome), which together give rise to the pangenome of such a population or species [175,176,177,208].

The numerous genome sequences allowed for a better understanding of the domestication processes of crops and animals and to follow some of the genetic changes that permitted domestication to occur [209]. However, the first fragmented data produced were insufficient to detect all or at least the most important genes associated with domestication syndromes [208]. However, they allow the implementation of GWAS to detect some domestication traits [210]. Since domestication reduces the genetic diversity of a taxon, often eliminating portions of the dispensable genome that contain genes involved in local adaptation, the use of wild relatives is crucial for generating a representative pangenome for a species [177]. Once a pangenome is generated, it can be used alongside whole-genome sequencing data to analyse the structural variants between and within populations, revealing novel loci involved in the development of domestication-related traits that would have remained hidden using only a single reference genome [211]. As sequencing technologies become cheaper, multiple pangenomes from different species of the same genus should eventually be combined to create a super-pangenome representing the entire genetic content available in a genus with one or more domesticated taxa, and it should include the diversity of all the wild relatives [175].

Wild plants can be regarded as reservoirs of useful genes [202], yet there is a vast array of plant species whose agricultural potential remains untapped [11]. Recent technological advances have opened a new approach to de novo domestication of wild plants as a viable solution for designing desired crops while maintaining food security and a more sustainable low-input agriculture. In these innovative fields, the potential application of CRISPR-like technologies for genome editing is very wide. The process of domesticating wild progenitors into edible crops is closely linked to the modification of developmental processes, and the steps that are needed to face the current challenges will equally require developmental modifications [212,213]. Therefore, studying the genetic basis of crop domestication is largely equivalent to studying aspects of plant development. Consequently, understanding the genetic and molecular mechanisms underlying developmental processes has great potential to further improve the performance of today’s major crops and determine routes for fast-tracking domestication of less developed crops [212]. Genes controlling plant development have been studied in multiple plant systems. This has provided deep insights into conserved genetic pathways controlling core developmental processes, including meristem identity, phase transitions, determinacy, stem elongation, and branching. These pathways control plant growth patterns and are fundamentally important in crop biology and agriculture.

There are recent examples of crops that have been targeted for rapid domestication [11]. Cape gooseberry or pichuberry (*Physalis peruviana*) was chosen for its growing popularity since it is highly nutritious and can be eaten as fresh fruit or used to make juice or jam. It is native to the Andean region of South America and has many “wild” characteristics that prevent it from being easily cultivated. Knowledge of the genes related to the improvement and domestication of the tomato, a distant relative of *P. peruviana*, has motivated scientists to identify similar genes in the undomesticated pichuberry that could be targeted for domestication. Gene editing of *P. peruviana’s* genetic ortholog of the tomato gene CLAVATA1 (SlCLV1), which controls meristem proliferation, gives rise to plants with narrow leaves and flowers with more organs. This offers a proof of principle for rapid, targeted domestication using gene editing. The cultivation of crops in urban environments may reduce the environmental impact of food production [37]. However, the lack of available land in cities and the need for rapid crop cycling to yield quickly and continuously mean that to date only lettuce and related ‘leafy green’ vegetables are cultivated in urban farms. New fruit varieties with architectures and yields suitable for urban farming have proven difficult to breed [37].

Despite all successes, gene editing approaches to the domestication of wild plants using domestication genes [11,202,207] should not be generally considered as simple one-time events in a single gene or generation. In contrast, de novo domestication literature and examples understand domestication as co-evolutionary interactions between plants and people that are complex, would require significant institutional and infrastructural investments, and can involve many disciplines. Gene editing approaches may help accelerate domestication and widespread cultivation of a new generation of soil-conserving and climate-smart crops.

## 16. Conclusions. Must I Be Cruel Only to Be Kind?

We, probably, can summarise that a revolutionary breakthrough in increasing the limit of crop yield by plant selection using targeted mutational changes in specific genes is unlikely. A replacement of inefficient photosynthetic machinery with more productive subunits for increasing the photosynthetic rate is attractive; however, there is typically a 20–30 year gap between the demonstration of innovative solutions at the experimental level and the provision of seeds to farmers [80]. Field trials involving genetically engineered plants are scarce worldwide and do not exist in Europe due to strict regulations. To date, most field trials involving genetically engineered varieties of rice, tomato, and other vegetables and crops are in Asia [214]. Thus, there seems to be little doubt that technology alone is powerless to feed the rapidly growing population.

In 1999, ecologist Peter Vitousek stated that “we are the first generation with tools to understand changes in the Earth’s systems caused by human activity, and the last with the opportunity to influence the course of many of these changes” (quoted from [215]). More than ever, humanity is changing the face of the planet at an increasing rate and conducting an unprecedented “environmental breakdown”. A century ago, only 15% of the Earth’s surface was modified by the direct effects of human activities [216]. This proportion has now grown to 87% of the ocean and 77% of the land [217]. This has led to a global collapse of biodiversity with an average 60% decline in populations of all vertebrate species and up to 83% for freshwater species between 1970 and 2014 as measured by the Living Planet Index [218]. Human activities currently threaten approximately 1 million species with extinction, with many others already extinct [215]. Considering this rapid loss of biodiversity, the world is now facing a sixth mass extinction [219]; the first to be caused by the species *Homo sapiens* [215].

Interestingly, this problem also involves crop cultivars that suffer from modern agricultural industry trends. Multiple cultivars are replaced by single “champions” for better productiveness and logistics. Over the decades, the pressure exerted by natural and artificial selection has progressively reduced the genetic diversity of many crops, including Italian durum wheat cultivars [220,221]. Modern industrial trends dictate regional crop specialisation, which removes all but a few crop species from traditional agricultural areas. For example, in a county-level study of individual crop land cover areas in the conterminous United States of America (U.S.A.) from 1840 to 2017, Michael Crossley and colleagues found a strong and abrupt spatial concentration of most crop types in recent years. For 13 of the 18 major crops, the widespread belts that characterised agriculture in the U.S.A. early in the 20th century collapsed, with spatial concentration increasing 15-fold after 2002. The number of counties producing each crop declined by up to 97% from 1940 to 2017, and their total area declined by up to 98% despite increasing total production. Consequently, a sharp decrease in crop types within counties occurred; in 1940, 88% of counties grew > 10 crops, but this figure was only 2% in 2017. Crossley et al. showed that declining crop diversity with increasing land area is a recent phenomenon, suggesting that the corresponding environmental effects in agriculturally dominated regions have fundamentally changed [222].

The discussion above was based on the need for resource growth, taking the inevitability of human population growth as a given. In 1798, Thomas Malthus wrote in his article “An Essay on the Principle of Population” that the rate of uncontrolled population growth always advances the growth of means of subsistence such that the exponential growth of an uncontrolled population is in contrast to the arithmetic growth of subsistence resources. The critics often referred to the “green revolution” as evidence that Malthus did not consider the technological factor in the production of foodstuffs. Indeed, as described above, during the past 30 years, the crop yield per unit time and land use for crop production has increased markedly [9].

In his Nobel speech, Norman Borlaug also warned: “The green revolution has won a temporary success in man’s war against hunger and deprivation; it has given man a breathing space. If fully implemented, the revolution can provide sufficient food for sustenance during the next three decades. However, the frightening power of human reproduction must also be curbed; otherwise, the success of the green revolution will be ephemeral only” [59]. Both Malthus and Borlaug drew attention to the alarming population growth, though neither could imagine the trajectory of agricultural development. Based on recent studies, Pete Smith and other scientists concluded that “Technology alone cannot provide food security in 2050. Food demand… will need to be managed if we are to continue to prove Malthus wrong into the future” [223]. However, these authors did not speak directly about the alarmingly expanding world population.

In the analysis of 12,640 research articles over the last 50 years, the authors of [10] identified three potential levers important for human population: total food production, per capita food demand, and population size. They reported a strong and increasing focus on feeding the world through increasing food production via technology, while the focus on reducing food demand through less intensive dietary patterns has remained constant and low. Population size has declined from being the dominant lever discussed in 1969 to the least researched in 2018.

Many consider that the Earth will soon reach its limits in terms of food supplies, natural resources, and pollution, and the world population will inevitably decline or even collapse due to successive uncontrollable crises. However, these assertions rarely factor in the internal constraints that shape population dynamics [224]. There is currently no country on Earth that meets the critical needs for human well-being while staying within the environmental planetary boundaries [10]. However, changing population size and age structure may have the most profound economic, social, and geopolitical impacts in many countries [225]. These problems can be dramatically enhanced by a lack of mineral resources [226].

Tampering with human population growth is a topic loaded with delicate moral issues. Those who accept the relevant scientific evidence are often accused of being genocidal, racist, anti-poor folks, anti-religion, and generally anti-human. In fact, those who accept the scientific imperative feel that they have a moral responsibility to be concerned about the future of mankind because it is increasingly apparent that without constraints on population growth, a sixth mass extinction [219] caused by *Homo sapiens* [215] seems to be inevitable [227]. For further information, see the gloomy but sobering predictions of overly optimistic individuals and governments [228].

Undeniably, the indefinite growth of both population and consumption is impossible on a planet with finite space and resources. In fact, meeting the UN Sustainable Development Goals is already regarded as requiring a lower world population growth [229]. However, the topic remained relatively poorly documented, and many people in scientific, policy, and public arenas continued to ignore or deny that population growth is an issue until relatively recently. In 2017, over 15,000 scientists from all over the world reported in a “warning to humanity” that population growth needed to be addressed; otherwise, all efforts to reach a sustainable future would be in vain. This outlines the need to bring population growth to the forefront of international concerns and overcome the taboos surrounding this question.

In 1998, Donella Meadows wrote: “facts... show Malthus to be not dead, not wrong, maybe not right either… Over another few decades, we will probably put old Malthus to rest at last. It’s up to us to decide whether he will rest triumphant or discredited”.

We would think he is at least not discredited, and we will have to think what can be done with our growing population and remember: “I must be cruel only to be kind” (Shakespeare, Hamlet).

## Figures and Tables

**Figure 1 plants-10-01667-f001:**
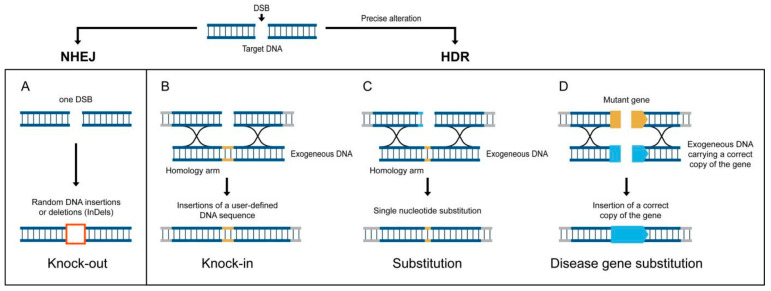
Genome-editing schemes with site-specific DNA nucleases. Double-stranded breaks (DSBs) induced by a nuclease (e.g., CRISPR/Cas9) at a specific site can be repaired by non-homologous end joining (NHEJ) or homology-directed recombination (HDR). (**A**) Repair by NHEJ usually results in the insertion or deletion of random base pairs causing gene knockout. (**B**) HDR with a donor DNA template with homologous arms can be exploited to achieve gene insertion and deletion, (**C**) to modify a gene by introducing precise nucleotide substitutions, and (**D**) inserting a correct gene copy instead of the mutant gene. DSB = double-stranded break.

**Figure 2 plants-10-01667-f002:**
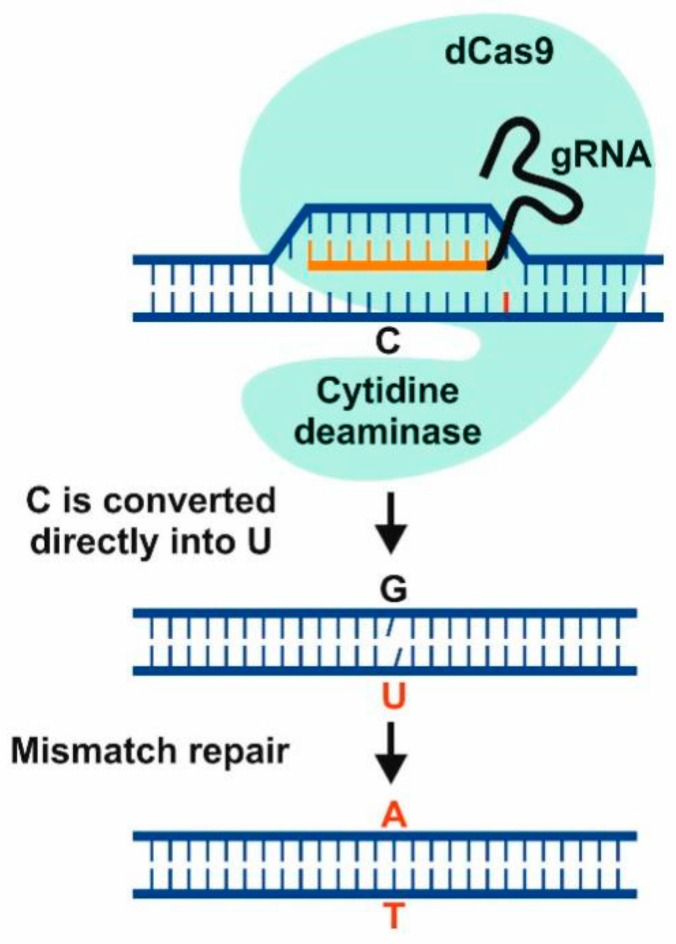
Base editing illustrated with cytidine deamination. In this type of base editing, cytidine deaminase fused with dCas9 targets the desired location. There is no DSB; C is converted directly into U on the free strand; and during mismatch repair, a C → T substitution can be created when the modified strand is used as a template [15].

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
