# Peer review of "Will Plant Genome Editing Play a Decisive Role in “Quantum-Leap” Improvements in Crop Yield to Feed an Increasing Global Human Population?"

_plants, 2021, doi:10.3390/plants10081667_

Round 1
Reviewer 1 Report
Манускрипт Буздина приводит обзор современных достижений и проблем в области геномного редактирования сельскохозяйственных растений для создания сортов растений с новыми свойствами в ответ на изменяющиеся условия существования человечества (увеличение народонаселения, изменение климатических и экологических условий в частности и под воздействием человека).
Данный обзор в большей степени фокусируется на постановке цивилизационных проблем и обсуждении возможностей новых технологий в генетике для их решения. Я полагаю, что такой взгляд на проблемы является важным, и позволяет осознать роль современной генетики в области сельскохозяйственных растений в решении глобальных задач. Мне кажется, что такие обзоры являются необходимы для более глубокого понимания направлений и результатов применения новейших генетических технологий.
Поэтому, я рекомендую публикацию данной работы.
Однако к тексту манускрипта у меня есть ряд замечаний:
Он содержит ряд технических недостатков в форматировании текста (см. второй сверху абзац на стр 14)
На странице 15 предложение «Theoretical limits on crop productivity» вероятно должно быть выделено как заголовок
На странице 19 год публикации работы Мальтуса 1798
Еще одно замечание: в некоторых местах авторы говорит о признаках урожайности растений упоминая колос, зерна и пр., что неявно подразумевает злаки. Однако для растений других видов признаки урожайности могут быть другими: плоды, корнеплоды, биомасса и пр. Это нужно каким-то образом уточнить или подкорректировать.
Buzdin's manuscript provides an overview of current advances and challenges in genomic editing of crops to create plant varieties with new properties in response to the changing conditions of human environment (population growth, changing climatic and environmental conditions in particular and under human influence).
This review focuses more on the statement of civilizational problems and a discussion of the possibilities of new technologies in genetics to solve them. I believe that this view is important and makes us aware of the role of modern crop genetics in addressing global challenges. It seems to me that such reviews are essential for a better understanding of the directions and results of the modern genetic technologies. Therefore, I recommend the publication of this work.
However, I have a number of comments on the text of the manuscript:
Several times the authors speak of plant yield traits by mentioning the ear, grains, etc., which implicitly implies cereal crops. However, for plants of other species the traits of yield may be different: fruits, roots, biomass, etc. This needs to be clarified or corrected in some way.
Text contains a number of technical deficiencies in the formatting of the text (see the second top paragraph on page 14):
On page 15 the sentence "Theoretical limits on crop productivity" should probably be highlighted as a heading
On page 19, the year of Malthus' work is 1798.
Author Response
We are grateful to the reviewer for the positive assessment of our work and valuable comments, with which we fully agree.
Below are the comments of the reviewer (italics) and our answers and changes in the text, which are also included in in the “Plants-1317288_corrections” file.
1.Text contains a number of technical deficiencies in the formatting of the text (see the second top paragraph on page 14.
We have tried to correct all the technical defects as indicated in the above indicated file “Plants-1317288_corrections”.
2.On page 15 the sentence "Theoretical limits on crop productivity" should probably be highlighted as a heading
In fact, it is on page 5. This is indeed the heading. We have corrected this in accordance with the original author's intention. This is now section 3. “Theoretical limits on crop productivity. A complex system cannot be predictably modified, but can be replaced by a functionally similar complex system”.
Accordingly, we have changed the numbering of subsequent sections, as indicated in the “Plants-1317288_corrections” file.
3.On page 19, the year of Malthus' work is 1798.
We corrected this typo.
4. Several times the authors speak of plant yield traits by mentioning the ear, grains, etc., which implicitly implies cereal crops. However, for plants of other species the traits of yield may be different: fruits, roots, biomass, etc. This needs to be clarified or corrected in some way.
We agree and have changed the text on page 7 in section 4, paragraph 2. In the original text it was: Crop yield is a QT [55] that is controlled by many plant genes with three main phenotypic components identified:
In the revised version it is: Crop yield is a QT [55] that is controlled by many plant genes. In wheat, for example, three main phenotypic yield components were identified
Reviewer 2 Report
In this paper, the authors have undertaken a very broad review of crop science and engineering.
While the review is largely well-written, there are areas that are incomplete and/or short relative to other sections and the voice varies in some sections. It would be helpful if the authors did a global edit to keep the voice consistent. Additionally each section does not need a conclusion (e.g., Section 8, page 13). The latter portion is better written and flows better than the beginning. Sections 10-13 clearly present the big ideas and ongoing questions. While much of the science has been the focus of previous reviews (which are cited) and can be read individually, this review brings together a broad range of topics pertaining to the food supply, challenges and solutions. I would remove the Shakespeare, Hamlet quote but it is probably a matter of personal preference. Specific suggested edits are listed below.
Suggested edits:
- While the authors introduce the issue of water in the introduction, it is currently not in the abstract but should be included (e.g., “water and land-use changes”).
- Gramene and GO ontology are referred to on page 3 then explained on page 15. It would be better to explain/define these at first use.
- Add detail to section 4. Nitrogen input
- Add detail in the first paragraph to 8. Genome editing technologies for “quantum-leap” improvements in yield-limited crops are ready, but where are targets? “There are two problems in the targeted genome editing: 1. Identification of target(s) which should be modified to achieve desirable phenotypic change and 2. Precise targeted genome modification” Add period after modification and add details or combine with paragraph 2.
- Page 3, remove space after Gene Ontology (GO).
- Page 3, give an example of “continual exploitation of crops tolerant to poor soil” such as breeding excess citric acid production to survive in acidic and soluble metal rich soils.
- Page 5, this sentence is not a complete paragraph and should be combined, extended, or deleted. “Theoretical limits on crop productivity. A complex system cannot be predictably modified, but can be replaced by functionally similar complex system”
- Page 7, as above, this sentence is not a complete paragraph and should be combined extended, or deleted. “However, scepticism remains whether increased photosynthetic capacity may increase food crop yields [104] (see below).”
- Page 8: NafY should be NifY
- Page 10, remove space after continued before period.
- On Page 12, should “2.2 × 3.9 x 10-8 per site” be 2.2 to 3.9 x 10-8 per site?
- Page 18, remove extra period after crops.
- Page 19, the date for Malthus should be 1798 not “1978”
- Section “Reference genomes and assessment of genomic variation” needs a number.
- There are some errors in reference formatting that will need to be addressed (e.g., references 14, 16).
Author Response
We are grateful to the reviewer for his very thorough analysis of our manuscript. We agree with the overwhelming majority of his comments, and only in two cases (suggestions 2 and 6, see below), we preferred to leave the situation unchanged although made certain adjustments to the text. We hope the reviewer will not mind, since in both cases we are not talking about fundamental changes.
Suggested edits:
1. While the authors introduce the issue of water in the introduction, it is currently not in the abstract but should be included (e.g., “water and land-use changes”).
Done
2. Gramene and GO ontology are referred to on page 3 then explained on page 15. It would be better to explain/define these at first use.
This remark (as well as others) demonstrates how carefully the reviewer analyzed the text. However, in this case, we would prefer to leave the layout unchanged. The reference on page 3 is in the Introduction, where a detailed explanation, in our opinion, would noticeably lengthen this section, which is already somewhat long. Therefore, we provide a more detailed explanation in section 12 on page 15 on phenotypic analysis of the role of genes, where these databases are widely used. However, in response to the reviewer's comment, and to make it easier for the reader to read the text, we added on page 3 a reference to section 12 and page 15. Now it is (see below, chapter 12, page 15). We really hope that the reviewer will agree with this reaction to his/her comment.
3. Add detail to section 4. Nitrogen input
Now it is section 5. We are grateful to the reviewer for drawing attention to the incompleteness of this section, which should be introducing the following chapters. This was caused by an accidental deletion from the original text of the concluding phrase: “In the following sections, we discuss some of the strategies intended to solve this critical issue” In the edited version, it is inserted.
4. Add detail in the first paragraph to 8. Genome editing technologies for “quantum-leap” improvements in yield-limited crops are ready, but where are targets? “There are two problems in the targeted genome editing: 1. Identification of target(s) which should be modified to achieve desirable phenotypic change and 2. Precise targeted genome modification” Add period after modification and add details or combine with paragraph 2.
Now this is chapter 9. We added the period. We chose the second option suggested by the reviewer and combined the paragraphs. The change is marked in the revised text.
5. Page 3, remove space after Gene Ontology (GO).
Done.
6. Page 3, to give an example of “continual exploitation of crops tolerant to poor soil” such as breeding excess citric acid production to survive in acidic and soluble metal rich soils.
We are thankful to the reviewer. Due to his/her remark we detected an inaccuracy that crept in during numerous edits of this text. The author of this idea Jonathan P. Lynch (reference 39) wrote that “the second Green Revolution will be based on crops tolerant of low soil fertility”. In our original text we cited this (and other) works in this way: “The suggestions vary starting from innovative urban agriculture development {Armanda, 2019 #347}{Kwon, 2020 #405}{Van-Erp, 2016 #185} and continuing with development of crops tolerant of low soil fertility{Lynch, 2007 #326}. In the text submitted to PLANTS this place is: “ continual exploitation of crops tolerant to poor soil” which incorrectly reflects the thought of JP Lynch.
Now we changed this place: to development of crops tolerant to poor soil”..
As to the reviewer’s suggestion to give an example of “continual exploitation….”, with all due respect to the opinion of the reviewer, we would prefer not to do this in the INTRODUCTION, especially since immediately after this phrase we write: “Due to the limited space, we will not consider the problems associated with the second and third green revolutions…”.
We would like to mention in this regard, that we are now preparing a review dedicated to strategies of local solutions to climate change where we are going to consider these problems in more detail and with relevant examples. We apologize for such a long answer to a short reviewer’s remark.
7. Page 5, this sentence is not a complete paragraph and should be combined, extended, or deleted. “Theoretical limits on crop productivity. A complex system cannot be predictably modified, but can be replaced by functionally similar complex system”
This is now the title of the chapter, as it was in the original and as noted by the reviewer
8. Page 7, as above, this sentence is not a complete paragraph and should be combined extended, or deleted. “However, scepticism remains whether increased photosynthetic capacity may increase food crop yields [104] (see below).”
We combined it with the previous paragraph
9. Page 8: NafY should be NifY
Done
10. Page 10, remove space after continued before period.
Done
11. On Page 12, should “2.2 × 3.9 x 10-8 per site” be 2.2 to 3.9 x 10-8 per site?
Certainly. Done
12. Page 18, remove extra period after crops.
Done
13. Page 19, the date for Malthus should be 1798 not “1978”
Done
14. Section “Reference genomes and assessment of genomic variation” needs a number.
Done. Now it has #11
15. There are some errors in reference formatting that will need to be addressed (e.g., references 14, 16).
We verified the references and made the necessary changes, both those indicated by the reviewer and others, related, for example, to the fact that in some cases the quoted articles were still in print and did not have pages.
Also, in the general part of his comments, the reviewer expressed an opinion: I would remove the Shakespeare, Hamlet quote but it is probably a matter of personal preference. We removed the reference to Shakespeare in the heading of the Conclusion.